# Region-specific myelin differences define behavioral consequences of chronic social defeat stress in mice

Valentina Bonnefil[1], Karen Dietz[2,3], Mario Amatruda[1], Maureen Wentling[1], Antonio V Aubry[4], Jeffrey L Dupree[5], Gary Temple[1], Hye-Jin Park[1], Nesha S Burghardt[4], Patrizia Casaccia[1,2,3], Jia Liu[1]*

[1]Advanced Science Research Center at the Graduate Center, Neuroscience Initiative, City University, New York, United States; [2]Department of Neuroscience, Icahn School of Medicine, New York, United States; [3]Friedman Brain Institute, Icahn School of Medicine, New York, United States; [4]Department of Psychology, Hunter College, City University, New York, United States; [5]Department of Anatomy and Neurobiology, Virginia Commonwealth University, Richmond, United States

**Abstract** Exposure to stress increases the risk of developing mood disorders. While a subset of individuals displays vulnerability to stress, others remain resilient, but the molecular basis for these behavioral differences is not well understood. Using a model of chronic social defeat stress, we identified region-specific differences in myelination between mice that displayed social avoidance behavior ('susceptible') and those who escaped the deleterious effect to stress ('resilient'). Myelin protein content in the nucleus accumbens was reduced in all mice exposed to stress, whereas decreased myelin thickness and internodal length were detected only in the medial prefrontal cortex (mPFC) of susceptible mice, with fewer mature oligodendrocytes and decreased heterochromatic histone marks. Focal demyelination in the mPFC was sufficient to decrease social preference, which was restored following new myelin formation. Together these data highlight the functional role of mPFC myelination as critical determinant of the avoidance response to traumatic social experiences.

**Editorial note:** This article has been through an editorial process in which the authors decide how to respond to the issues raised during peer review. The Reviewing Editor's assessment is that all the issues have been addressed (see decision letter).

DOI: https://doi.org/10.7554/eLife.40855.001

*For correspondence:
Jia.Liu@asrc.cuny.edu

**Competing interests:** The authors declare that no competing interests exist.

## Introduction

Exposure to stress increases the risk of developing affective disorders such as depression and post-traumatic stress disorder. While stress leads to maladaptive behavioral responses in a subset of humans, others are capable of coping and remain resilient. Differences in the behavioral response to stress can also be detected in experimental mouse models, thereby highlighting the degree of conservation of this response. However, the cellular and molecular basis underlying resilience or susceptibility to negative experiences remains poorly defined.

We and others have previously reported that animal models of psychosocial stressors, such as social isolation (*Liu et al., 2012*; *Liu et al., 2016*; *Makinodan et al., 2016*; *Makinodan et al., 2017*; *Makinodan et al., 2012*), chronic social defeat stress (CSDS) (*Cathomas et al., 2019*; *Lehmann et al., 2017*), and chronic variable stress (*Liu et al., 2018*), lead to transcriptional, translational, or ultrastructural changes in oligodendrocytes and myelination. Here we tested the hypothesis that myelinating glia serves a causal role in behavioral susceptibility or resilience following stress

**eLife digest** High levels of stress do not have the same effect on everybody: some individuals can show resilience and recover quickly, while other struggle to cope. Scientists have started to investigate how these differences may find their origin in biological processes, mainly by focusing on the role of neurons. However, neurons represent only one type of brain cells, and there is increasing evidence that interactions between neuronal and non-neuronal cells play an important role in the response to stress.

Oligodendrocytes are a common type of non-neuronal cells which shield and feed nerve cells. In particular, their membrane constitutes the myelin sheath, a protective coating that insulates neurons and allows them to better communicate with each other using electric signals.

Bonnefil et al. explored whether differences in oligodendrocytes could affect how mice responded to social stress. The rodents were exposed to repeated attacks from an aggressive mouse five minutes a day for ten days. After this period, 'susceptible' mice then avoided future contact with any other mice, while resilient animals remained interested in socializing.

Comparing the brain areas of resilient and susceptible mice revealed differences in the oligodendrocytes of the medial prefrontal cortex, the part of the brain that controls emotions and thinking. Susceptible animals had fewer mature oligodendrocytes and their neurons were covered in thinner and shorter segments of myelin sheaths. There was also evidence that, in these animals, the genes that regulate the maturation of oligodendrocytes were more likely to be switched off. Taken together, these results may suggest that, in certain animals, social stress disrupts the genetic program that controls how oligodendrocytes develop, potentially leading to neurons communicating less well.

To explore whether reduced amounts of myelin could be linked to decreased social behavior, Bonnefil et al. then damaged the myelin in the medial prefrontal cortex in another group of rodents. The mice were then less willing to interact with other animals until new sheaths had formed.

The results by Bonnefil et al. undercover how changes in non-neuronal cells can at least in part explain differences in the way individuals respond to stress. Ultimately, this knowledge may be useful to design new strategies to foster resilience.

DOI: https://doi.org/10.7554/eLife.40855.002

exposure. We examined social behaviors, ultrastructural changes in myelination as well as epigenetic modifications in oligodendrocytes in brain regions that have been implicated in depressive-like behavior after a well-established social defeat paradigm (*Berton et al., 2006*; *Golden et al., 2011*; *Hodes et al., 2014*; *Krishnan et al., 2007*; *Vialou et al., 2010*). We also provide mechanistic insights into the region-specific differences between the phenotypes, which we attributed to defective oligodendrocyte progenitor differentiation. To provide direct causal evidence, we carried out focal demyelination in the medial prefrontal cortex and showed aversive social behavior in animals undergoing demyelination and a resolution of the behavioral effect consequent to new myelin formation. Together, we suggest the functional role of region-specific myelination in determining depressive-like social behavior.

## Results and discussion

### Chronic social defeat stress causes region-specific changes in myelination

We adopted a mouse model of chronic social defeat stress (CSDS) (*Golden et al., 2011*), in which mice were exposed to an aggressor challenge for 10 days (*Figure 1A*) and tested for social behavior afterwards. While some mice showed signs of social withdrawal, characterized by reduced social interaction time when a conspecific mouse is present and reduced social interaction ratio (i.e. susceptible mice), a subset escaped this deleterious consequence (i.e. resilient mice), and were virtually indistinguishable from the control group, which were not exposed to any aggressors (*Figure 1B–C*).

Next, we sought to determine whether there was any myelination difference between susceptible and resilient mice. We focused our analysis on the nucleus accumbens (NAc) and the medial

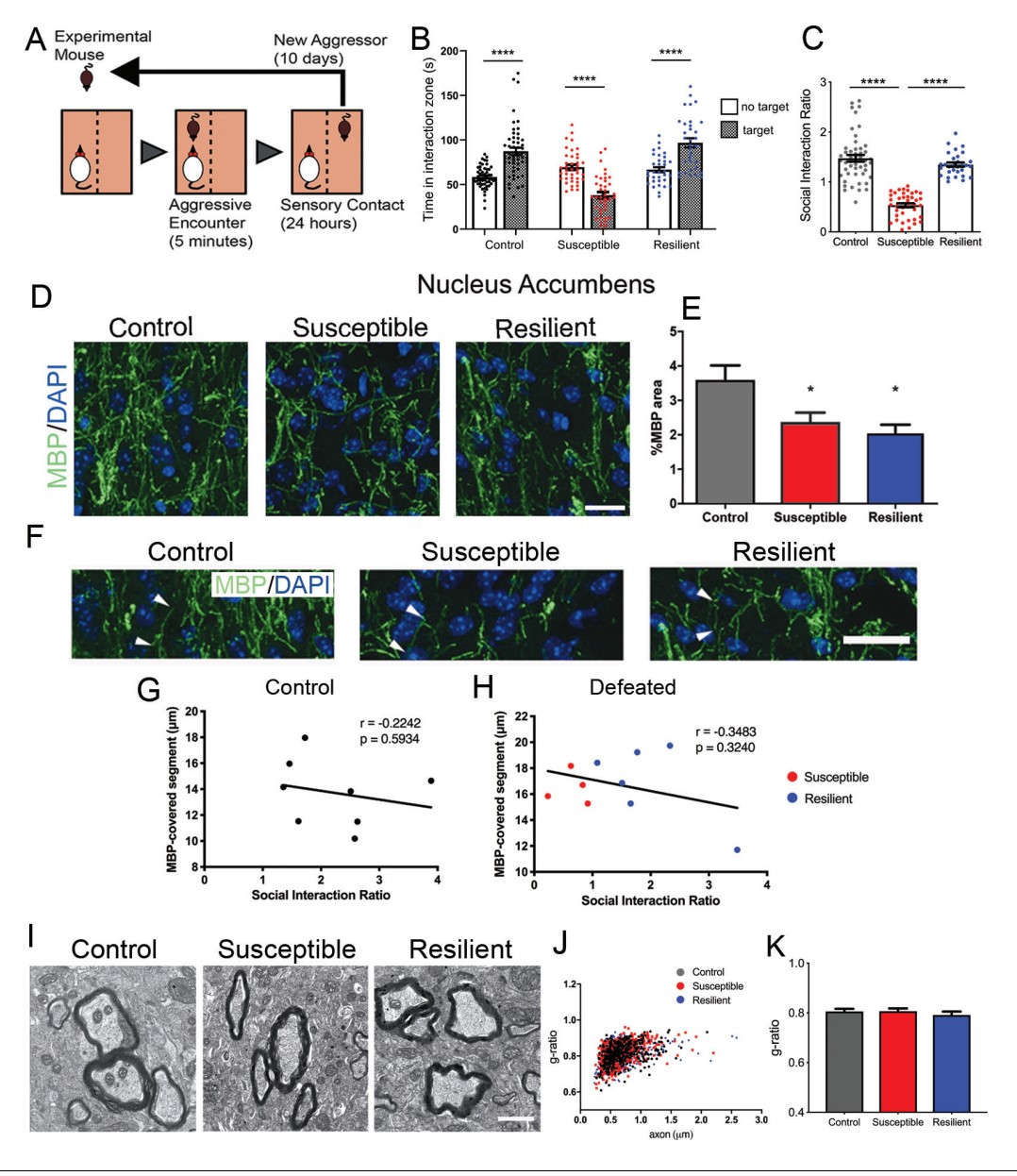

**Figure 1.** Effect of aggressive social encounters on myelination in the nucleus accumbens (NAc) of mice which showed two behaviorally distinct phenotypes following chronic social defeat stress (CSDS). (**A**) The experimental paradigm for CSDS. (**B–C**) Mice susceptible to CSDS spent less time interacting with a conspecific mouse than the control group or resilient mice, as shown in (**B**) total time spent in the interaction zone when there is a conspecific mouse present and in (**C**) social interaction ratio defined by time spent in the interaction zone when a conspecific mouse present divided by a conspecific mouse absent. Control, n = 52; susceptible, *n* = 39; resilient, n = 33; ****, p<0.0001 by one-way ANOVA followed by Tukey's post hoc test. (**D–E**) Representative confocal images and quantifications showing immunohistochemistry of myelin basic protein (MBP) counterstained with DAPI. Scale bar = 28 μm. n = 3 mice per group, two 20x images per animal; susceptible vs. control, p=0.0447; resilient vs. control, p=0.0109 by one-way ANOVA followed by Tukey's *post hoc* test. (**F**) Representative confocal images showing MBP-covered myelinated segments. Arrowheads point to one MBP-covered myelinated segment. Scale car = 19 μm. (**G–H**) Pearson correlation coefficients showed non-significant correlation of MBP-covered segment length and social interaction ratio in control (**G**) or defeated (**H**) mice, control, 8 *x-y* pairs. r = −0.2242, p=0.5932. defeated, 10 x-y pairs, r = −0.3483, p=0.3240. control, n = 8 mice, susceptible, n = 4 mice, resilient, n = 6 mice, 1–2 20x images per mouse; (**I**) Representative electron microscopy images (scale bar = 1 μm) and (**J–K**) scatter plot and quantification of g-ratio; control, n = 5 mice; susceptible, n = 7 mice; resilient, n = 5 mice.

*Figure 1 continued on next page*

*Figure 1 continued*

DOI: https://doi.org/10.7554/eLife.40855.003

The following source data is available for figure 1:

**Source data 1.** Source data for social interaction behavior following chronic social defeat stress and myelin content in the nucleus accumbens.
DOI: https://doi.org/10.7554/eLife.40855.004

prefrontal cortex (mPFC), two brain regions shown to play a critical role in determining stress responses (*Heshmati et al., 2018*; *Han and Nestler, 2017*) and displaying myelin transcriptional or structural impairment after a stressful experience (*Liu et al., 2012*; *Liu et al., 2016*; *Makinodan et al., 2016*; *Lehmann et al., 2017*; *Liu et al., 2018*; *Zhang et al., 2016*). In the NAc, a significant reduction of myelin basic protein (MBP) was detected in all defeated mice, regardless of their behavioral responses (control, 3.6 ± 0.4%; susceptible, 2.4 ± 0.3%; resilient 2.0 ± 0.3%; *Figure 1D–E*). However, no significant differences were detected in the length of myelinated segments measured by MBP immunoreactivity (*Figure 1F–H*) or in myelin thickness (*Figure 1I–K*) among groups. Pearson coefficients correlation showed no significant correlation between the length of MBP-covered segments and social interaction ratio in either control or defeated group (control, r = −0.2242, p=0.5934, defeated, r = −0.3483, p-0.3240, *Figure 1G–H*). Altogether, these results suggest that myelination in the NAc uniformly responds to stress and does not distinguish susceptibility and resilience following CSDS.

In contrast, the mPFC displayed a unique myelination phenotype following CSDS. While the levels of MBP did not significantly differ between susceptible and resilient mice (control, 6.2 ± 0.5%; susceptible, 6.1 ± 0.9%; resilient, 5.3 ± 0.8%; *Figure 2A–B*), the length of myelinated segments indicated by MBP immunoreactivity showed a significant positive correlation with social interaction in defeated mice (*Figure 2C–E*). Importantly, such correlation was not detected in the control (unstressed) group, suggesting that changes in the length of myelinated segments represent an adaptive response to the social defeat stress. To more accurately quantify internodal length, we conducted immunohistochemical analysis using antibodies specific for the contactin-associated protein (Caspr), which marks the paranodal regions (*Figure 2F*). Also in this case, a significant positive correlation between internodal length and social interaction ratio was detected only in the defeated mice, with shorter internodal lengths identified in susceptible mice (*Figure 2G–H*). Myelin was also thinner in the susceptible - but not in the resilient – mice, compared to controls (*Figure 2I–K*). Therefore, region-specific myelination differences in the mPFC could -at least in part- explain the behavioral differences between susceptible and resilient mice in response to stress.

## Different oligodendrocyte populations in the mPFC of susceptible and resilient mice

To determine whether reduced myelin content in the mPFC of susceptible mice was limited to the internodal length, we further performed a detailed quantitative immunohistochemical analysis on oligodendrocyte lineage cells. No significant difference in the overall number of OLIG2+ cells was detected (*Figure 3A,C*), thereby ruling out decreased survival of oligodendrocyte lineage cells in response to stress. However, compared to resilient and controls, the susceptible mice were characterized by a significantly higher number of NG2+ progenitor cells (control, 39.2 ± 2.3 mm$^{-2}$; susceptible, 55.3 ± 4.4 mm$^{-2}$; resilient, 25.8 ± 4.8 mm$^{-2}$; *Figure 3A,C*), and lower number of CC1+ mature oligodendrocytes (control, 91.9 ± 5.7 mm$^{-2}$; susceptible, 62.7 ± 5.2 mm$^{-2}$; resilient, 86.1 ± 5.8 mm$^{-2}$; *Figure 3B–C*). Consistent with defective differentiation of NG2+ cells in the mPFC of susceptible mice, a reduction of the histone modification marks associated with differentiation (H3K9me3) was also detected (pixel/area: control, 1373.6 ± 113.3; susceptible, 695.3 ± 127.9; resilient, 1186.0 ± 106.0; *Figure 3D*). Together these data suggest that social stress might have at least two main effects on oligodendrocyte lineage cells in the mPFC: it promotes myelin remodeling resulting in shorter internodal length and fewer wraps and possibly impairs in the epigenetic program of oligodendrocyte progenitor differentiation, resulting in fewer differentiated oligodendrocytes.

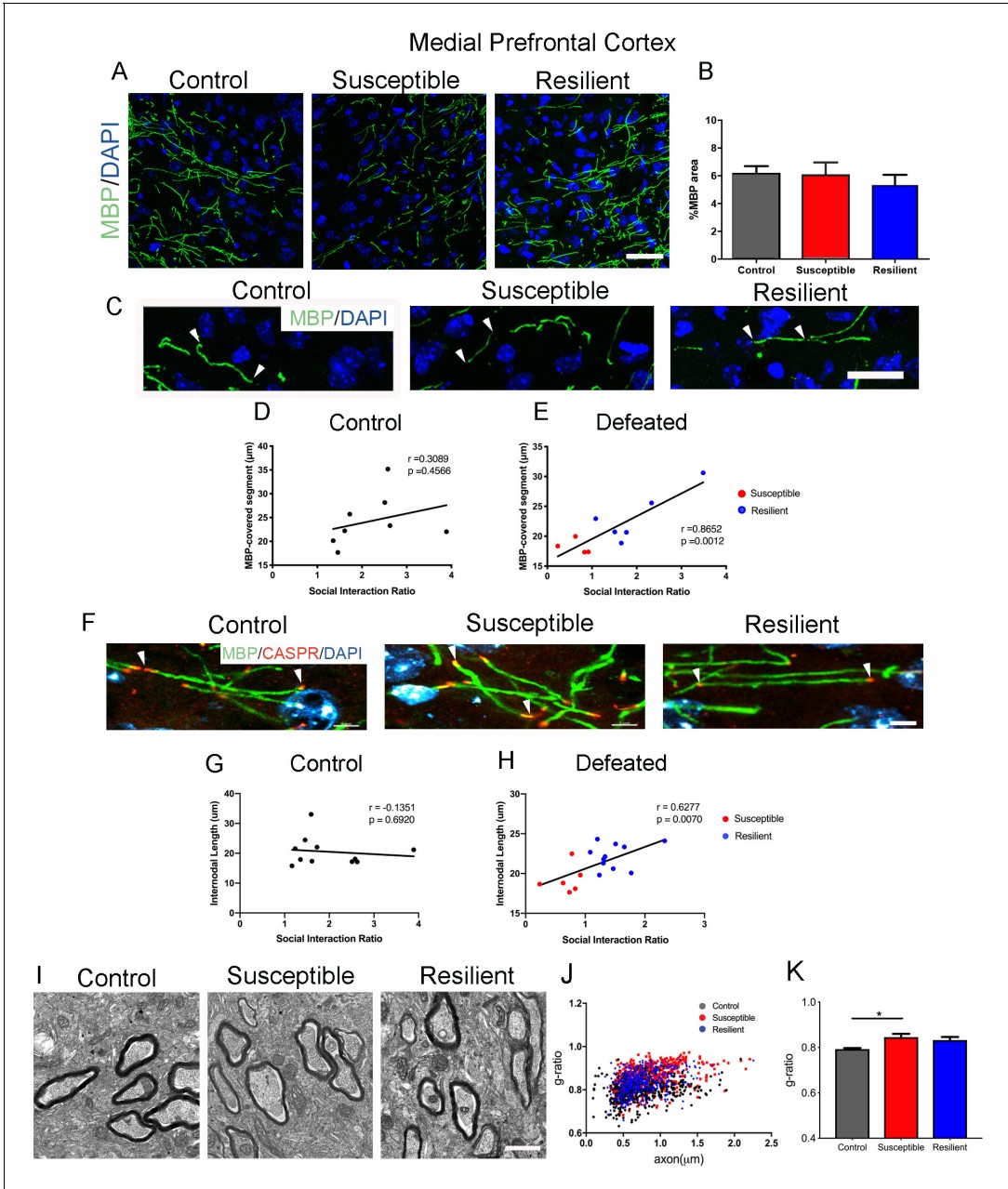

**Figure 2.** Myelination in the medial prefrontal cortex (mPFC) distinguished resilient from susceptible mice following stress. (A–B) Representative confocal images and quantifications of myelin basic protein (MBP) counterstained with DAPI. Scale bar = 30 μm. n = 3 mice per group. Four 20x images taken per mouse (C) Representative confocal images showing MBP-covered myelinated segments. Arrowheads point to one continuous MBP-covered myelinated segment. Scale bar = 17 μm. (D–E) Pearson correlation coefficients showed significant correlation of MBP-covered segment length with social interaction ratio only in defeated (E) mice, but not in control (D), control, 8 x-y pairs, n = 8 mice, defeated 10 x-y pairs, susceptible, n = 4 mice, resilient, n = 6 mice, four 20x images were taken per mouse (F) Representative confocal images showing internodal segment marked by CASPR (Red) and MBP (Green). Arrowheads point to one internode. Scale bar = 5 μm. (G–H) Pearson correlation coefficients showed significant correlation of internodal length with social interaction ratio only in defeated (H) mice, but not in control (G), control, 11 x-y pairs, n = 11 mice, defeated 17 x-y pairs, susceptible, n = 6 mice, resilient, n = 11 mice, four-six 63x images taken per mouse. (I) Representative electron microscopy images, scale bar = 1 μm. (J–K) Scatter plot and quantification of g-ratio in the mPFC; control, n = 5 mice; susceptible, n = 7 mice; resilient, n = 5 mice; susceptible vs. control, p=0.0264 by one-way ANOVA followed by Tukey's *post hoc* test.

DOI: https://doi.org/10.7554/eLife.40855.005

The following source data is available for figure 2:

**Source data 1.** Source Data for myelin content in the medial prefrontal cortex in social defeat mice.
DOI: https://doi.org/10.7554/eLife.40855.006

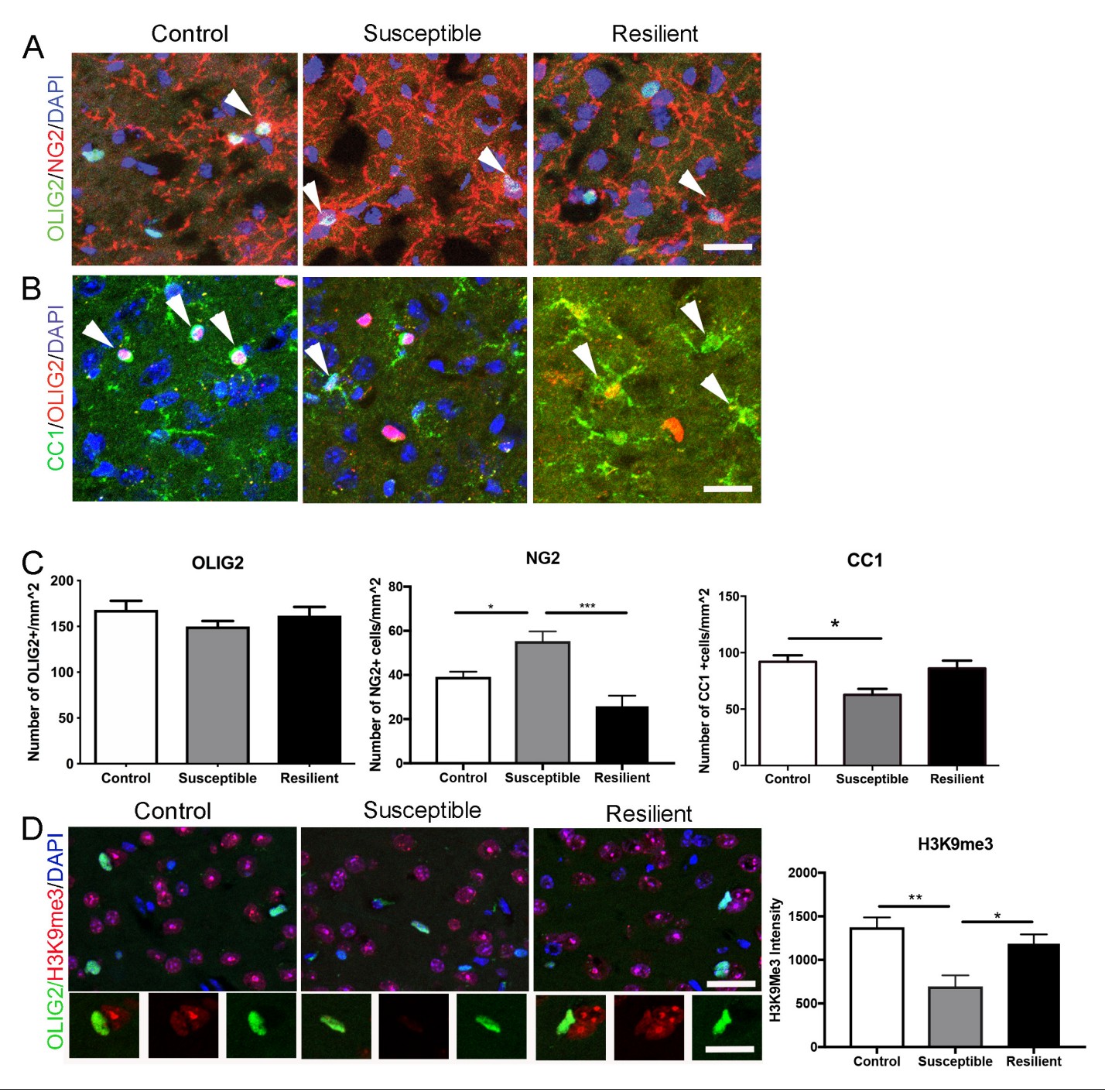

**Figure 3.** Impaired oligodendrocyte differentiation was associated with reduced repressive histone methylation marks in the mPFC of susceptible mice. (A–B) Representative confocal images of cells positive for OLIG2, NG2, and CC1 in the mPFC. DAPI was used as counterstain of nuclei. Scale bar = 25 µm. (C) quantification of OLIG2+ (n = 3 mice per group and 3–4 20x images taken per mouse), NG2+ cells (n = 3 mice per group and four images taken per mouse, *p=0.019, ***p<0.0001 by one-way ANOVA followed by Tukey's *post hoc* test) and CC1+ cells (control, n = 8 mice, susceptible, n = 3 mice, resilient n = 7 mice, 3–4 20x images taken per mouse; *p=0.0195 by one-way ANOVA followed by Tukey's *post hoc* test). (D) Representative confocal images and quantifications (E) of mean intensity of repressive histone mark H3K9me3 (Red) in OLIG2+ (Green) cells. control, n = 3 mice, susceptible, n = 2 mice, resilient, n = 5 mice, 4 20x images taken per mouse, 50–100 OLIG2+ cells were counted per image **p=0.0038, *p=0.0234 by one-way ANOVA followed by Tukey's *post hoc* test. Data are mean ± S.E.M. Scale bar = 20 µm.

DOI: https://doi.org/10.7554/eLife.40855.007

The following source data is available for figure 3:

*Figure 3 continued on next page*

*Figure 3 continued*

**Source data 1.** Source data for the number of oligodendrocyte lineage cells and intensity of a repressive histone mark, H3K9me3, in the medial prefrontal cortex.

DOI: https://doi.org/10.7554/eLife.40855.008

## Focal demyelination in the mPFC leads to social avoidance behavior

The data above suggested an interesting correlation between myelination in the mPFC and social avoidance behavior in the susceptible mice. To test the causality of this finding, we induced myelin loss by focal injection of lysolecithin (LPC) into the mPFC and asked whether this manipulation would be sufficient to induce behavioral changes. LPC injection is a well characterized model of toxic demyelination, with early myelin loss (detectable one week after injection) followed by spontaneous repair, due to the formation of new myelin by newly differentiated oligodendrocytes (occurring three weeks after injection) (*Jeffery and Blakemore, 1995*). We reasoned that behavioral differences detected in mice at these two time points after LPC injection would support a causal link between social preference performance and myelin content in the mPFC (*Figure 4A*). Indeed, the kinetics of demyelination and remyelination after LPC injection was validated by the detection of reduced MBP immunoreactivity at the 7dpi followed by spontaneous recovery of immunoreactivity by 21dpi (*Figure 4B*). At the early time point (7dpi), LPC-injected mice displayed reduced social preference behavior compared to saline-injected controls (*Figure 4C*). This difference in social interaction behavior was no longer detectable after 3 weeks (*Figure 4D*), when myelination recovered to normal level (*Figure 4E*). Therefore, we conclude that myelin content in the mPFC is a critical determinant of social behavior.

Altogether, our study reveals region-specific epigenetic dysregulation of oligodendrocyte progenitor differentiation and subsequent defective adult myelination as maladaptive mechanisms

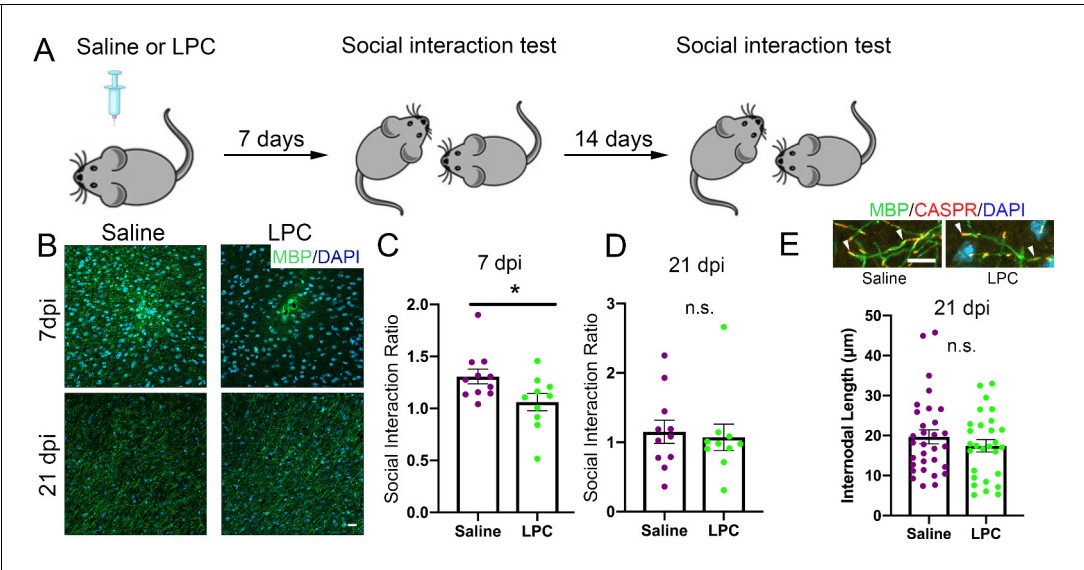

**Figure 4.** Focal demyelination in the mPFC reduced social preference behavior. (**A**) The experimental paradigm for lysolecithin (LPC) injection and behavioral testing. (**B**) Representative confocal images showing reduced MBP immunointensity at seven dpi followed by a spontaneous restoration at 21 dpi. (**C**) Mice received LPC displayed reduced social preference behavior at seven dpi as quantified by social interaction ratio. (**D**) Restoration of normal social interaction behavior at 21 dpi as quantified by social interaction ratio. Saline, n = 11 mice; LPC, n = 10 mice *, p<0.05 by unpaired t-test. Data are mean ± S.E.M. (**E**) Representative confocal images of immunohistochemistry of MBP (Green) and CASPR (Red) and scatter plots of internodal length at 21dpi. Counterstained with DAPI. Saline n = 3 mice, LPC, n = 5 mice. 2–4 63x images taken per mouse.

DOI: https://doi.org/10.7554/eLife.40855.009

The following source data is available for figure 4:

**Source data 1.** Source data for social interaction behavior and internodal length following lysolecithin-induced focal demyelination.

DOI: https://doi.org/10.7554/eLife.40855.010

occurring only in susceptible mice after exposure to repeated social stress. We have previously reported that myelination defects were detected in socially isolated adult mice, prior to the appearance of social avoidance behavior (*Liu et al., 2012*). Here, we show that social avoidance behavior can be detected after chronic social defeat stress as well as after focal demyelination in the mPFC, and could therefore be caused by hypomyelination. Furthermore, promoting myelination has been shown to rescue depressive-like behavior in socially isolated mice (*Liu et al., 2016*). On the same note, normal social behavior was restored following the spontaneously occurring remyelination in LPC-injected mice. However, it is important to note that, social stress did not induce a toxic effect on myelin, whereas LPC did. No cellular toxicity was detected in the mPFC of susceptible mice or in mice undergoing social isolation (*Liu et al., 2012*). In contrast, we detected fewer mature oligodendrocytes and more progenitors lacking epigenetic marks of differentiation, suggesting an altered epigenetic program. For this reason, we interpret the lower myelin content in the mPFC of susceptible mice as resulting from impaired oligodendrocyte progenitor differentiation, possibly as maladaptive response to social stress. While our data support an inefficient production of new myelin, the detection of a positive correlation between intermodal length and social avoidance behavior, suggests that reorganization of paranodal loops could also be affected. Indeed, shorter internodal length, consequent to impaired myelin formation, has been previously shown to decrease nerve conduction in the optic nerve (*Etxeberria et al., 2016*). It is, therefore, conceivable that the reduced length of myelinated segments detected in the mPFC of susceptible mice may result in slower conduction and functionally result in the characteristic social avoidance behavior in response to the social stress.

Finally, we suggest that new myelin is formed in the mPFC of resilient mice as an adaptive mechanism to the repeated episodes of aggression. It is conceivable that the formation of new myelin in resilient mice could favor the establishment of neuronal circuits allowing the escape of negative impact following traumatic stress (*Krishnan et al., 2007*; *Fagundes et al., 2013*; *Fenster et al., 2018*; *Ménard et al., 2017*; *Russo et al., 2012*), as oligodendrocytes are known to regulate conduction speed and play a crucial role in synchronizing neuronal networks (*Saab and Nave, 2017*). This explanation is in agreement with the increasing evidence from mice and squirrel monkeys, which suggests stress resilience may arise from active coping strategies, rather than a passive response, defined as lack of adaptive response (*Russo et al., 2012*; *Lyons et al., 2009*).

The molecular basis for resilience has been studied extensively in the context of neuronal cells, the immune and neuroendocrine systems (*Ménard et al., 2017*; *Russo et al., 2012*). Here we proposed an alternative, although not mutually exclusive explanation involving myelinating glia. One possibility for new myelin formation as a coping strategy is associated with increased neuronal activity in the resilient mice, as reported by a greater degree of FosB, or ΔFosB expression in glutamatergic neurons of mPFC of resilient mice following social defeat (*Covington et al., 2010*; *Lehmann and Herkenham, 2011*). Optogenetic stimulation of mPFC has been shown to help resilience phenotype in social defeated mice (*Covington et al., 2010*). Although not characterized in the previous study (*Covington et al., 2010*), optogenetic stimulation has been shown to promote oligodendrogliogenesis and new myelin formation (*Gibson et al., 2014*). An alternative mechanism could involve inflammatory cytokines, such as interleukin-6 (IL-6). IL-6 has been identified as a major cytokine that contributes to the development of depression in human (*Dowlati et al., 2010*; *Erta et al., 2012*). In animal models of stress, systemic IL-6, was the only differentially regulated cytokine that distinguished resilient mice from susceptible and control mice (*Hodes et al., 2014*). Although systemic changes of IL-6 could not account for the region-specific differences in myelination in susceptible and resilient mice, it is known that IL-6 can be produced by neurons, astrocytes, microglia or endothelial cells in the central nervous system (*Erta et al., 2012*). Several transcriptomic studies suggest that oligodendrocyte progenitors express IL-6 receptors (*Zeisel et al., 2015*; *Zhang et al., 2014*). Therefore, it is intriguing to think that IL-6 could be up-regulated in a region-specific pattern with the ability to impact oligodendrocyte progenitor differentiation and new myelin formation in specific regions of the adult brain.

Overall this study extends our knowledge on the functional role of adult myelination by providing a mechanism for adaptation to social stress encounters, which ultimately result in the expression of resilience.

# Materials and methods

**Key resources table**

| Reagent type (species) or resource | Designation | Source or reference | Identifiers | Additional information |
|---|---|---|---|---|
| *M. musculus* (C57Bl/6J) | mouse | Jackson Laboratory | RRID:IMSR_JAX:000664 | |
| *M. musculus* (CD-1) | mouse | Charles River | RRID:IMSR_CRL:22 | Retired breeder |
| Antibody | Mouse monoclonal anti-MBP | Covance | Cat # SMI99 RRID:AB_2564741 | IHC (1:500) |
| Antibody | Rabbit polyclonal anti-Caspr | Abcam | Cat# ab34151, RRID:AB_869934 | IHC (1:100) |
| Antibody | Mouse monoclonal anti-OLIG2 | Millipore | Cat# MABN50, RRID:AB_10807410 | IHC (1:200) |
| Antibody | Rabbit polyclonal anti-OLIG2 | Abcam | Cat# ab81093, RRID:AB_1640746 | IHC (1:200) |
| Antibody | Rabbit anti-H3K9me3 | Abcam | Cat# ab8898, RRID:AB_306848 | IHC (1:100) |
| Antibody | Rabbit polyclonal anti-NG2 | Millipore | Cat# AB5320, RRID:AB_91789 | IHC (1:200) |
| Antibody | Mouse monoclonal anti-APC | EMDMillipore | Cat# OP80, RRID:AB_2057371 | IHC (1:100) |
| chemical compound, drug | DAPI | Thermofisher | Cat# D1306, RRID:AB_2629482 | IHC (1:10000) |
| chemical compound, drug | l-α-lysophosphatidylcholine | Sigma-Aldrich | Cat# L4129 | |
| software, algorithm | ImageJ | | RRID:SCR_003070 | |
| software, algorithm | Ethovision XT | Noldus | RRID:SCR_000441 | |
| software, algorithm | Graphpad Prism 8 | | RRID:SCR_002798 | |

## Animals

All experimental C57Bl/6J male mice (7 weeks) were obtained from the Jackson Laboratory (Bar Harbor, Maine) and allowed one-week acclimation prior to the start of experiment. Retired male CD1 breeders used as the aggressors were obtained from Charles River (Wilmington, Massachusetts). All mice were maintained in a temperature- and humidity-controlled facility on a 12 hr light-dark cycle with food and water ad libitum. All procedures were carried out in accordance with the Institutional Animal Care and Use Committee guidelines of the Icahn School of Medicine at Mount Sinai, Hunter College and Advanced Science Research Center at City University of New York.

## Chronic social defeat stress.

Chronic social defeat stress was performed as previously published (*Berton et al., 2006*; *Golden et al., 2011*; *Krishnan et al., 2007*; *Vialou et al., 2010*; *Wilkinson et al., 2009*) with slight modification. Briefly, male C57 mice were exposed to a novel aggressive CD1 male mouse for 5 min/day, after which the mice were separated by a Plexiglas barrier that allows for sensory contact without further physical interaction. Control mice were housed two animals/cage under the same conditions as their experimental counterparts but without the presence of an aggressive CD1 mouse. Twenty-four hours after the last of 10 daily defeat or control episodes, mice were evaluated in a social interaction test during the light cycle, as previously described (*Liu et al., 2012*), then one-way ANOVA tests were performed to assess statistical differences and assess social avoidance. Social interaction ratio was calculated by dividing the time spent in the interaction zone when a conspecific mouse is present by no subject present in the enclosure area. Defeated mice with a social interaction ratio below one are defined as 'susceptible', while those with a social interaction ratio above one are defined as 'resilient'.

## Electron microscopy

Mice were processed for standard electron microscopy (EM) analysis as previously described (*Liu et al., 2012*). Briefly, the mounted section was trimmed to encompass a $1 \ \mu m^2$ region of layers 4–6 of the PFC, thin sectioned at 90 nm, stained with uranyl acetate and lead citrate, and mounted on 200 mesh copper grids. Ten images at 10,000X were collected per mouse using a transmission electron microscope JEOL JEM 1400Plus equipped with a Gatan CCD camera. g-ratios were determined by dividing the diameter of the axon by the diameter of the entire myelinated fiber. ImageJ was used to measure both axon caliber and myelin fiber diameter for a minimum of 100 myelinated axons per mouse. All analyses were performed blind to the experimental conditions. One-way ANOVA tests were performed to assess statistical differences.

## Immunohistochemistry

Mice were anesthetized and then perfused, cryopreserved, embedded, and sectioned as previously described (*Liu et al., 2012*). Immunohistochemistry was performed as previously described (*Liu et al., 2012*) with primary antibodies against trimethylated histone 3 lysine 9 (H3K9me3, 1:100; ab8898, Abcam), CC1 (1:100; OP80, Calbiochem), myelin basic protein (MBP, 1:500; SMI99, Covance), OLIG2 (1:200, ab81093, Abcam), NG2 (1:200; AB5320, EMD Millipore) or Caspr (1:100, ab34151, Abcam). Stained sections were visualized using confocal microscopy (LSM800 Meta confocal laser scanning microscope, Carl Zeiss Micro-Imaging). For NG2, CC1, OLIG2 cell counts, and H3K9me3 intensity quantifications, 4–6 20x fields were taken per mouse. For MBP-covered segments and internodal length marked by Caspr, 4–6 fields were taken per mouse followed by quantifications using ImageJ. One-way ANOVA tests were performed to assess statistical differences. For correlation of internodal length with social interaction ratio, data normality was determined using D'Agostino and Person test in GraphPad Prism 8. Pearson correlation coefficients were calculated if data passed normality test.

## Stereotaxic surgery for lysolecithin injection

While under deep anesthesia induced by inhaled isoflurane, experimental C57BL/6J mice were surgically injected with 1 µl 1% lysolecithin (l-α-lysophosphatidylcholine, Sigma-Aldrich) dissolved in saline, or saline as sham control, bilaterally to the medial prefrontal cortex using a pulled capillary glass pipet at the following stereotaxic coordinates: anterioposterior,+1.5 mm; mediolateral from bregma, 0.5 mm; and dorsoventral-below the surface of the dura, 1.5 mm. The needle was left in place for an additional 2 min to avoid back flow of the lysolecithin or saline. Muscle and skin incisions were sutured with gut and nylon sutures, respectively. To reduce postoperative pain after recovery from anesthesia, animals received a subcutaneous injection of buprenorphine (1.0 mg/kg). Animals were monitored closely following surgery and were tested with social interaction tests at 7- and 21 days post injection.

## Acknowledgements

We thank Drs. Eric Nestler, Scott Russo and Rosemary Bagot for the help with animal behavioral assessments. We thank the Epigenetics Core Facility and the Rodent Behavioral Suite at CUNY Advanced Science Research Center for technical help. This work is supported by National Institute of Neurological Disorders and Stroke (2R37NS042925-10, R01NS52738 to PC), NIH-NINDS Center Core Grant 5P30 NS047463, NIH-NCI Cancer Center Grant P30 CAO16059 to JLD, National Institute on Minority Health and Health Disparities (NIMHD) of the NIH (MD007599) to NSB, and City University of New York PSC-CUNY awards to JL. We apologize to our colleagues whose work we did not cite due to limited space.

## Additional information

### Funding

| Funder | Grant reference number | Author |
|---|---|---|
| National Institute of Neurological Disorders and Stroke | 2R37NS042925-10 | Patrizia Casaccia |
| National Institute of Neurological Disorders and Stroke | R01NS52738 | Patrizia Casaccia |
| National Institute of Neurological Disorders and Stroke | CenterCore Grant 5P30 NS047463 | Jeffrey L Dupree |
| National Cancer Institute | Cancer Center Grant P30 CAO16059 | Jeffrey L Dupree |
| National Institute on Minority Health and Health Disparities | MD007599 | Nesha S Burghardt |
| City University of New York | | Jia Liu |
| National Institute of Mental Health | R21MH114182 | Nesha S Burghardt |

The funders had no role in study design, data collection and interpretation, or the decision to submit the work for publication.

### Author contributions

Valentina Bonnefil, Data curation, Formal analysis, Investigation, Methodology, Writing—original draft, Writing—review and editing; Karen Dietz, Hye-Jin Park, Methodology, Acquisition of data and data analysis; Mario Amatruda, Maureen Wentling, Antonio V Aubry, Gary Temple, Methodology, Acquisition of data; Jeffrey L Dupree, Data curation, Formal analysis, Methodology; Nesha S Burghardt, Supervision, Methodology, Acquisition of data; Patrizia Casaccia, Conceptualization, Funding acquisition, Writing—review and editing; Jia Liu, Conceptualization, Data curation, Formal analysis, Supervision, Funding acquisition, Validation, Investigation, Methodology, Writing—original draft, Project administration, Writing—review and editing

### Author ORCIDs

Valentina Bonnefil https://orcid.org/0000-0001-6651-9612
Mario Amatruda https://orcid.org/0000-0002-5407-238X
Antonio V Aubry https://orcid.org/0000-0001-8604-356X
Nesha S Burghardt https://orcid.org/0000-0002-5415-1141
Jia Liu https://orcid.org/0000-0001-6274-2710

### Ethics

Animal experimentation: This study was performed in strict accordance with the recommendations in the Guide for the Care and Use of Laboratory Animals of the National Institutes of Health. All procedures were carried out in accordance with the Institutional Animal Care and Use Committee guidelines of the Icahn School of Medicine at Mount Sinai (Protocol number 08-0676), Hunter College (Protocol number NB-stress 6/18-T3 and NB fear 9/19-02) and Advanced Science Research Center at City University of New York (Protocol number 2018-8).

### Decision letter and Author response

Decision letter https://doi.org/10.7554/eLife.40855.013
Author response https://doi.org/10.7554/eLife.40855.014

## Additional files

### Supplementary files

• Transparent reporting form
DOI: https://doi.org/10.7554/eLife.40855.011

### Data availability

All data generated or analyzed during this study are included in the manuscript. Source data files have been provided.

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
