## [Decision Letter]

[**Editorial note:** This article has been through an editorial process in which the authors decide how to respond to the issues raised during peer review. The Reviewing Editor's assessment is that all the issues have been addressed.]

Thank you for submitting your article "Region-specific myelin differences define behavioral consequences of chronic social defeat stress in mice" for consideration by *eLife*. Your article has been reviewed by three peer reviewers, and the evaluation has been overseen by a Reviewing Editor and Catherine Dulac as the Senior Editor. The following individual involved in review of your submission has agreed to reveal his identity: Mathias Schmidt (Reviewer #3).

The Reviewing Editor has highlighted the concerns that require revision and/or responses, and we have included the separate reviews below for your consideration. If you have any questions, please do not hesitate to contact us.

Summary:

Studying the resilience or susceptibility to stress with a focus on the role of myelin is important and timely, given the recent advances in our understanding myelin plasticity and its impact on higher brain functions. Overall, this is an exciting study demonstrating such a role of myelin in social behaviour and maladaptive social behavior. The authors present evidence that resilience to stress requires maintenance of healthy myelin in the medial prefrontal cortex (mPFC), and that stress susceptible subjects exhibit impaired oligodendrocyte differentiation dynamics following the stressful condition specifically in mPFC.

Major concerns:

The authors interpret the data shown in Figure 1B as proof that exposure to the aggressive mouse changed the behavior of the test mice. However, they do not show that the social behaviors were changed by the experience of aggression since they do not compare the social behavior before and after exposure.

The question emerges whether PFC myelination affects the performance in social interactions and/or is affected itself by social defeat. The authors need to show that there is no correlation between the characteristics of PFC myelin and social interactions in control mice (this appears critical as Liu et al. previously reported a link between PFC myelin and social interactions).

Throughout the text, authors should give exact definition of "n" and state when it is number of mice, number of axons etc. Also, "imaging volumes" should be defined.

Throughout the study, authors use MBP immunohistochemistry alone to measure the myelin internode length. However, MBP alone may not differentiate the difference between an internode and an axon leaving the obrvational field in z-axis. Co-staining with Caspr should clarify that distinction. Rather than re-analyze all of the internode data this way, the authors could demonstrate in a sample group that Caspr co-staining validates the original internode quantifications.

At 21dpi, authors show a recovery of social interaction with the recovery of myelination. Does the recovery of internode length also occur?

The authors should plot individual data points in the bar graphs to better estimate group size and distribution of the data. How exactly were resilient and vulnerable animals defined?

The first experiment included a large number of mice, while all follow-up analyses only used very few animals per group (e.g. 3 or 4). How were these animals selected from the respective groups? What was the social avoidance behavior of the selected mice?

The differences in histone modification are interesting, but only correlative. The authors should avoid claiming a causal relationship with myelination or stress susceptibility.

One is not convinced by the conclusions drawn from the LPC experiment. At 7 days following treatment, the authors observed a reduction in MBP levels. However, MBP levels were not significantly different between stress resilient and susceptible mice. The manipulation does therefore not reflect the stress-induced situation, even though a similar behavioral phenotype was observed.

The authors tend to overstate their conclusions, as all observations are correlational and no experiments were performed that would indicate a causal relationship between the observed differential myelination phenotype of resilient and susceptible animals with their social behavior or epigenetic regulation.

Separate reviews (please respond to each point):

*Reviewer #1:*

The study by Bonnefil et al. explores "the cellular and molecular basis underlying resilience or susceptibility to negative experiences", focusing on the role of myelin. This is an important and timely question, given the recent advances in the understanding of CNS myelin plasticity and its impact on behavior. However, several of the key experiments have major issues and the results do not convincingly support the author's conclusions.

Main concerns:

The authors use a well-established social defeat paradigm to differentiate between mice that are susceptible and resilient based on how they behave on a social interaction test after exposure to an aggressive mouse. Later, they use this difference to explore the role of myelination on social interactions. I'm concerned with the basic assumptions of this approach. The authors interpret the data shown in Figure 1B as proof that exposure to the aggressive mouse changed the behavior of the test mice. However, there are two significant problems. 1) They do not really show that the social behaviors were changed by the experience of aggression since they do not compare the social behavior before and after exposure. 2) It is not clear that there is a significant difference between the naïve (control) group and all the exposed mice (resilient and susceptible groups together). A lack of difference would suggest that the important question is whether PFC myelination impacts the performance in social interactions rather than have anything to do with the social defeat. The authors need to show there is no correlation between the characteristics of PFC myelin and social interactions in control mice before justifying looking at the effects of the defeat paradigm. This is critical as Liu and Casaccia and others previously reported a link between PFC myelin and social interactions. Moreover, the observation that lysolecithin-induced focal demyelination in the mPFC leads to social avoidance by itself further suggests that the social defeat might be irrelevant.

The characterization of the differences between the susceptible and resilient mice at the cellular levels (Figure 3) has several problems.

1) It is my understanding that Olig2 is a transcription factor and therefore should be detected in the nucleus. However, the olig2 staining seems to extend beyond the nucleus. I would have expected to colocalize with the H3K9me3, which for sure should be nuclear. The authors need to use a nuclear dye to prove that the stainings are in the right cellular compartments.

2) The olig2 staining in panel A is very different from the one shown in panel D both in density and quality, raising doubts about the results.

3) It is not clear that measuring the intensity of the H3K9me3 is the right variable to compare. It seems that% of olig2+ cells with H3K9me3 staining is necessary as well.

4) How can the density of CC1+ and olig2+ cells be the same? it is my understanding that CC1 marks mature oligodendrocytes, while olig2 also labels precursors.

Based on the results shown in Figure 3A, the authors conclude that the exposure doesn't affect survival of cells of the oligodendrocyte lineage, yet they found a reduction in CC1+ cells (3C). Do they propose that mature oligodendrocyte revert to a progenitor state?

In the Introduction (second paragraph) the authors claim that this study provides " mechanistic insights into the region-specific differences between the phenotypes, which we attribute to defective epigenetic regulation of oligodendrocyte progenitor differentiation." Given the issues with Figure 3, I don't believe that the results support this conclusion.

In the third paragraph of the Results section the authors mention that susceptible mice display significantly shorter myelinated segments. How was the internode length measured?

In the Discussion, the authors state that "new myelin is formed in the mPFC of resilient mice as an adaptive mechanism to the repeated episodes of aggression". What data do they use to reach this conclusion ?

Other concerns:

In the Introduction the statement that "multiple" brain regions were evaluated is not appropriate given that only NAc and mPFC were described in the study.

Even though no significant correlation was seen between myelination in the NAc and social interaction in mice after chronic social defeat stress, the manuscript would benefit with a discussion on what mechanisms would possibly underlie the region-specific differences observed in the NAc and mPFC. How is the profile of expression of OLIG2, NG2, CC1 and H3K9me3 in the NAc in susceptible and resilient mice? Is the NAc somehow protected from the effects of defeat stress on myelination?

How was the proper sample size determined in each protocol of the study? The figure legend 1 indicates that 66 controls, 67 susceptible and 50 resilient mice were used in the chronic social defeat stress paradigm. On the other hand, the immunohistochemistry data that strongly support the assumptions of the study were obtained from only 2 to 3 mice per group, according to the Materials and methods section. Such small n for the histology studies does not generate confidence in the results.

All graphs in the results, particularly for the behavior assessment, should show the distribution of each individual sample, as points in the bar graph.

For rigor and reproducibility, the time of the day for the behavioral assessments and the anesthetic used for the stereotaxic surgery need to be added to the Material and methods section.

How did the authors confirm that lysolecithin injection hit the desired site in the mPFC? What is the success rate of the stereotaxic surgery?

Were the same mice tested for social interaction at 7- and 21- days post lysolecithin injection? Is it possible that those mice present any adaptation to the experimental protocol that could bias the results of social interaction?

The images in the Figure 4B do not clearly show an increase of MBP expression in mice 21 days after LPC injection, compared to 7 days post treatment. Higher magnification images might better represent the remyelination process after 21 days.

Because IL-6 circulating levels or mRNA expression were not assessed, the discussion on IL-6 role in resilience and susceptibility to stress seems unwarranted.

*Reviewer #2:*

This study investigates the relationship between myelin structure and chronic social defeat stress to pinpoint the cellular basis underlying susceptibility or resilience to negative social experiences. A well-established chronic social defeat paradigm is used to identify the different behavioural responses to stress. Two distinct phenotypes emerge as the susceptible mouse group shows social withdrawal and resilient mouse group continues to exhibit normal social interaction.

These two group of mice also show differences in myelin and in oligodendrocyte cell linage differences in the mPFC region but not the NAc. Authors investigate the causal relationship between myelination and behavioural susceptibility to stress using a focal demyelination model. Demyelination of mPFC causes a decrease in the social interaction, similarly to stress exposure, and remyelination rescues that behavioural phenotype.

Overall, this is an exciting study demonstrating the role of myelin in social behaviour and maladaptive social behaviour following stress conditions in functionally relevant brain regions. The authors present evidence that resilience to stress requires maintenance of healthy myelination in mPFC, and that a susceptible subject exhibits impaired oligodendrocyte differentiation dynamics following the stressful condition specifically in mPFC, implicating disordered mPFC myelination in susceptibility to stress-induced social withdrawal. Together, the manuscript represents an important contribution to the field and lends compelling support for the concept that dysmyelination can play important roles in vulnerability to stress, and mood disorders.

I have the following suggestions for improvement to better support the interesting claims:

Regarding statistical rigor: Throughout the text, authors should give exact definition of "n" and state when it is number of mice, number of axons etc. Also, "imaging volumes" should be defined.Figure 1A. The social defeat paradigm could be explained better by including the social interaction test that is performed in the end of chronic social defeat setting. This will make it clearer for the reader and it would be more coherent for Figure 4, and also help explain Figure 1B which shows the time spent in interaction zone that is measured during social interaction test.Figure 1B. Including social interaction ratio as well will make it more consistent with the rest of the figure.Including more information on control group animals in the text would make it easier to understand the differences between mice groups. For example, "control group, mice that were not exposed to any aggressor" would be helpful in understanding the comparisons.Throughout the study, authors use MBP immunohistochemistry alone to measure the myelin internode length. However, MBP alone may not differentiate the difference between an internode and an axon going into the field in z-axis. Co-staining with caspr should clarify that distinction. Rather than re-analyze all of the internode data this way, the authors could demonstrate in a sample group that caspr co-staining validates the original internode quantifications.Including a scatter plot (g-ratio vs axon calibre) is a more informative way to present g-ratio data.What does% MBP+ area measurement represent? Unaffected N.Ac. shows reduction in susceptible and resilient mice, but the affected mPFC show no difference for% MBP+ area. Maybe there are fewer myelinated axons in N.Ac for susceptible and resilient mice groups, does the EM data give any suggestion towards this?Figure 4. Regarding social interaction test, presentation of time spent in the interaction zone or distance travelled in the presence and absence of a target mouse would show that subject mouse is fine to leave the interaction zone (i.e. motor abilities). In other words, unpacking the social interaction ratio.Figure 4. At 21dpi, authors show that there is recovery of social interaction with the recovery of myelination in mPFC. Does the recovery of internode length also occur? Or MBP area, to reconcile with the initial mPFC phenotype they show in Figure 2?

Minor Comments:

The reduction in H3k9me3 immunostaining in Olig2+ cells in mPFC is consistent with the observed decreased OPC differentiation, but it seems an overstatement to claim "region-specific epigenetic dysregulation of oligodendrocyte differentiation", There is not at present evidence that the lack of mPFC oligodendrogenesis is primarily due to epigenetic dysregulation…the reduced H3K9me3 could alternatively simply reflect the loss of differentiation signal from the environment. Wording should be modified.Regarding wording, some softening of claims should be made in the Discussion. For example, "social avoidance behavior can be detected after chronic social defeat stress as well as after focal demyelination in the mPFC, and is therefore caused by hypomyelination." Would be better reading, "…and therefore can be caused by hypomyelination"

*Reviewer #3:*

In the current manuscript the authors examine the effects of 10 days of chronic social defeat (CSD), a well-known chronic stress model, on myelin differences in two brain regions, the nucleus accumbens and the medial prefrontal cortex (mPFC). When the CDS animals were stratified by their social avoidance behavior in susceptible (high social avoidance) and resilient (low social avoidance, comparable to un-stressed controls), the authors observed that only in the mPFC susceptible animals were characterized by shorter myelinated segments and decreased myelin thickness. In parallel, susceptible mice had a reduced number of mature oligodendrocytes and decreased H3K9 methylation marks. Finally, the authors chemically and transiently induced demyelination in the mPFC and observed social avoidance behavior only under conditions of demyelination, not following spontaneous remyelination. The data are potentially interesting. At the same time I have a number of questions and concerns, which I believe the authors should address:

a) It would be helpful if the authors would plot individual data points in the bar graphs to better estimate group size and distribution of the data. How exactly where resilient and vulnerable animals defined?

b) The first experiment included a large number of mice, while all follow-up analyses only used very few animals per group (e.g. 3 or 4). How were these animals selected from the respective groups? What was the social avoidance behavior of the selected mice? Specifically I am worried that if only extremes of a group were selected that the results would not be representative of the whole group.

c) For the correlation analyses in Figure 1F and 2F: Are animals from all 3 groups included? The data would be more convincing if a correlation of intermodal length and social interaction ratio would be present within experimental groups or at least within the CSD group.

d) The differences in histone modification are interesting, but only correlative. The authors should avoid claiming a causal relationship with myelination or stress susceptibility.

e) I am not convinced by the conclusions the authors draw from the LPC experiment. At 7 days following treatment, the authors observed a reduction in MBP levels. However, MBP levels were not significantly different between stress resilient and susceptible mice. The manipulation does therefore not reflect the stress-induced situation, even though a similar behavioral phenotype was observed.

f) How specific is the social avoidance phenotype induced by focal demyelination using LPC? The authors would need to show that the animals are not generally impaired and healthy.

g) While the final experiment parallels the behavioral effect in stress susceptible mice (but see the issue with MBP levels), it does not indicate causality. For that, the authors would need to show that a prevention of demyelination would increase stress resilience following CSD.

Overall, I think the data are novel and interesting. However, the authors overstate their conclusions, as all observations are correlational and no experiments were performed that would indicate a causal relationship between the observed differential myelination phenotype of resilient and susceptible animals with their social behavior or epigenetic regulation.

---

## [Author Response]

Major concerns:The authors interpret the data shown in Figure 1B as proof that exposure to the aggressive mouse changed the behavior of the test mice. However, they do not show that the social behaviors were changed by the experience of aggression since they do not compare the social behavior before and after exposure.

We respectfully note that the reviewer’s comment addresses the validity of the model of social defeat. While our manuscript has adopted this methodology, we kindly refer previous literature on the characterization of this model in mice (Golden et al., 2011). We also note that a large literature supports social stress as modulator of social behavior in mice. A PubMed search querying for ‘Social Defeat Stress In Mice’ returned over 600 studies conducted between 1984 and 2019, with over 400 references within the past 5 years. Several previous publications (e.g. Mul et al., Neuropsychopharmacology, 2018; Muir et al., Neuropsychopharmacology, 2018), have reported that depressive-like behaviors -such as sucrose preference and social interaction – were indistinguishable between susceptible and resilient mice prior to the aggressor encounter, and were discriminated only after the social defeat stress. Importantly, mice in the two groups were carefully controlled for sex, age, and genetic background at the beginning of each experiment. Although one could posit that sporadic differences in social behavior might have pre-existed the encounter with the aggressor, it is important to note that differences in social avoidance behavior were only detected in susceptible mice after the aggressor encounter and not in the control group, which was not exposed to any aggressor.

The question emerges whether PFC myelination affects the performance in social interactions and/or is affected itself by social defeat. The authors need to show that there is no correlation between the characteristics of PFC myelin and social interactions in control mice (this appears critical as Liu et al. previously reported a link between PFC myelin and social interactions).

We appreciate the reviewer’s comment. In agreement with this suggestion, we have revised Figure 2 and included additional panels (Figure 2D, 2E, 2G, and 2H) showing lack of correlation between myelin and behavior in the control group and the existence of a correlation between internodal length and social behavior only in mice after the aggressor encounter.

Throughout the text, authors should give exact definition of "n" and state when it is number of mice, number of axons etc. Also, "imaging volumes" should be defined.

We thank the reviewer for the rigorous review. We have now modified the figure legends to include a clear definition of number of mice, imaging fields analyzed or number of counted cells. In addition, “imaging volumes” has been replaced by “20x images” or “63x images”.

Throughout the study, authors use MBP immunohistochemistry alone to measure the myelin internode length. However, MBP alone may not differentiate the difference between an internode and an axon leaving the obrvational field in z-axis. Co-staining with Caspr should clarify that distinction. Rather than re-analyze all of the internode data this way, the authors could demonstrate in a sample group that Caspr co-staining validates the original internode quantifications.

This is a valid concern, which led us to perform additional experiments using immunohistochemistry with antibodies for Caspr and MBP, in order to carefully determine the internodal length in co-stained axonal segments. The new results mirror our previous measurements using MBP and are now presented in revised Figure 2 and Figure 4.

At 21dpi, authors show a recovery of social interaction with the recovery of myelination. Does the recovery of internode length also occur?

In agreement with the reviewer’s request, we have now performed additional experiments using Caspr and MBP immunohistochemistry and behavior and demonstrated a recovery of the internodal length at 21 days post injection.

The authors should plot individual data points in the bar graphs to better estimate group size and distribution of the data. How exactly were resilient and vulnerable animals defined?

In agreement with the reviewer’s suggestion we have now replaced the bar graphs in multiple panels with individual data points, to reflect data distribution. Susceptible and resilient mice were defined on the basis of the social interaction test results (Golden et al., 2011). Social interaction ratio was determined by the time spent in the interaction zone with a conspecific mouse divided by the time spent in the absence of a mouse. A mouse with social interaction ratio below 1 would be classified as “susceptible”, whereas a mouse with a ratio above 1 would be classified as “resilient”. A detailed description on the classification of mice as susceptible and resilient is also included in the Materials and methods section.

The first experiment included a large number of mice, while all follow-up analyses only used very few animals per group (e.g. 3 or 4). How were these animals selected from the respective groups? What was the social avoidance behavior of the selected mice?

Due to the nature of the large variability in behavioral experiments and the small amount of tissue available in each region of interest for each molecular analysis, we had to use a large number of mice and repeat the experiments in several cohorts. As mentioned, resilient and susceptible groups were based on their social interaction ratio. The number of mice in the follow-up analysis was based on the experimental outcome, with resilient and susceptible mice from the same cohort being analyzed in each independent determination.

The differences in histone modification are interesting, but only correlative. The authors should avoid claiming a causal relationship with myelination or stress susceptibility.

While we appreciate the reviewer’s comment, we kindly note that previous work demonstrated the necessity of specific histone modifications for oligodendrocyte lineage progression (Liu et al., 2015). Although the differences in histone modifications may not be causally linked to stress susceptibility, it is conceivable to interpret decreased histone methylation and defective lineage progression in the susceptible mice as being causally related. Nevertheless, we have attempted to better clarify this concept in the revised text.

One is not convinced by the conclusions drawn from the LPC experiment. At 7 days following treatment, the authors observed a reduction in MBP levels. However, MBP levels were not significantly different between stress resilient and susceptible mice. The manipulation does therefore not reflect the stress-induced situation, even though a similar behavioral phenotype was observed.

We agree with the reviewer. In response to this comment, we have performed additional experiments to measure the internodal length after LPC injection. As shown in the revised Figure 4 we detect a clear correlation between this parameter and social behavior after stress.

The authors tend to overstate their conclusions, as all observations are correlational and no experiments were performed that would indicate a causal relationship between the observed differential myelination phenotype of resilient and susceptible animals with their social behavior or epigenetic regulation.

We have revised the manuscript to avoid overstating conclusions from our experiments.

Separate reviews (please respond to each point):

Reviewer #1:

The study by Bonnefil et al. explores "the cellular and molecular basis underlying resilience or susceptibility to negative experiences", focusing on the role of myelin. This is an important and timely question, given the recent advances in the understanding of CNS myelin plasticity and its impact on behavior. However, several of the key experiments have major issues and the results do not convincingly support the author's conclusions.Main concerns:The authors use a well-established social defeat paradigm to differentiate between mice that are susceptible and resilient based on how they behave on a social interaction test after exposure to an aggressive mouse. […] The authors need to show there is no correlation between the characteristics of PFC myelin and social interactions in control mice before justifying looking at the effects of the defeat paradigm. This is critical as Liu and Casaccia and others previously reported a link between PFC myelin and social interactions. Moreover, the observation that lysolecithin-induced focal demyelination in the mPFC leads to social avoidance by itself further suggests that the social defeat might be irrelevant.

These concerns have been addressed above, in the response to the editor.

The characterization of the differences between the susceptible and resilient mice at the cellular levels (Figure 3) has several problems. 1) It is my understanding that Olig2 is a transcription factor and therefore should be detected in the nucleus. However, the olig2 staining seems to extend beyond the nucleus. I would have expected to colocalize with the H3K9me3, which for sure should be nuclear. The authors need to use a nuclear dye to prove that the stainings are in the right cellular compartments.

We have included a nuclear DAPI staining to demonstrate that Olig2 is clearly located inside the nucleus.

2) The olig2 staining in panel A is very different from the one shown in panel D both in density and quality, raising doubts about the results.

This is likely due to a suboptimal rendering of the figure. We have revised the figure to include distinct image selection, that we hope will address the reviewer’s concern.

3) It is not clear that measuring the intensity of the H3K9me3 is the right variable to compare. It seems that% of olig2+ cells with H3K9me3 staining is necessary as well.

While the reviewer’s suggestion is very appropriate in case of marker expression, H3K9me3 is a histone modification which defines heterochromatin and, as such, present in all the cells. The intensity measurement takes into account the fact that H3K9me3 is a histone mark which is deposited during the differentiation of progenitors into oligodendrocytes. For this reason, one would expect that progenitors are characterized by lower levels of intensity for H3K9me3 staining than differentiated oligodendrocytes.

4) How can the density of CC1+ and olig2+ cells be the same? it is my understanding that CC1 marks mature oligodendrocytes, while olig2 also labels precursors.

The similar density of CC1+ and Olig2+ cells is due to the fact that Olig2 persists in mature cells. We have carefully re-evaluated the cell density of all populations. While the majority of Olig2+ cells are also CC1+, the density of Olig2+ cells was slightly higher than CC1+, as noted by the reviewer.

Based on the results shown in Figure 3A, the authors conclude that the exposure doesn't affect survival of cells of the oligodendrocyte lineage, yet they found a reduction in CC1+ cells (3C). Do they propose that mature oligodendrocyte revert to a progenitor state?

It is important to distinguish between overall reduction in number of CC1+ in the presence or absence of increased NG2+ cells. The latter is shown in Figure 3C, and suggests that the reduction of CC1+ cells is due to halted differentiation of oligodendrocyte progenitor cells.

In the Introduction (second paragraph) the authors claim that this study provides " mechanistic insights into the region-specific differences between the phenotypes, which we attribute to defective epigenetic regulation of oligodendrocyte progenitor differentiation." Given the issues with Figure 3, I don't believe that the results support this conclusion.

We thank the reviewers for allowing us to clarify the interpretation of the data. We have previously demonstrated a critical role of histone methylation in regulating oligodendrocyte lineage progression (Liu et al., 2015). In this manuscript we show the concomitance of impaired oligodendrocyte progenitor differentiation and decreased histone methylation in susceptible, compared to resilient mice. By inference, we suggest that the detected alterations of oligodendrocyte lineage progression in susceptible mice could be due to aberrant histone modifications. In agreement with the reviewer, we have toned down the interpretation of the experimental results and clarified the statement.

In the third paragraph of the Results section the authors mention that susceptible mice display significantly shorter myelinated segments. How was the internode length measured?

The internodal length was measured by quantifying MBP+ immunoreactive segments, flanked by Caspr+ immunoreactivity.

In the Discussion, the authors state that "new myelin is formed in the mPFC of resilient mice as an adaptive mechanism to the repeated episodes of aggression". What data do they use to reach this conclusion?

It is technically very challenging to directly demonstrate new myelin formation in a behavioral setting experiment and we have reached the conclusion based in suggestive cumulative assessment of the experimental results. We found that resilient mice displayed longer internodal length and were characterized by higher numbers of differentiated oligodendrocytes, higher levels of histone marks and increased myelin thickness, compared to susceptible mice. We reasoned that these differences between the two groups could either be explained by the “escape” from an injury in resilient mice or by an active compensatory mechanism to the social stress. We tend to favor the latter. In addition, the positive correlation between internodal length and social interaction behavior detected only in mice exposed to the aggressor, but not in controls, further suggested the existence of an active response to social stress. This explanation is in agreement with the increasing evidence from mice and squirrel monkeys, which suggests stress resilience may arise from active coping strategies.

Other concerns:In the Introduction the statement that "multiple" brain regions were evaluated is not appropriate given that only NAc and mPFC were described in the study.

We have revised the text to replace “multiple” with “two brain regions”.

Even though no significant correlation was seen between myelination in the NAc and social interaction in mice after chronic social defeat stress, the manuscript would benefit with a discussion on what mechanisms would possibly underlie the region-specific differences observed in the NAc and mPFC. How is the profile of expression of OLIG2, NG2, CC1 and H3K9me3 in the NAc in susceptible and resilient mice? Is the NAc somehow protected from the effects of defeat stress on myelination?

This is an interesting point and would need additional experiments to address. However, we did examine the profile of oligodendrocyte lineages cells in the NAc but did not include in the manuscript due to the limit of numbers of illustrations as a short report. We did not detect significant differences in OLIG2+ in the NAc. The density of CC1+ cells showed a trend of increase in both susceptible and resilient mice but did not reach statistical significance. We have included a discussion on potential mechanisms accounting for the region-specific differences observed in the NAc and mPFC, which we attributed to potential differences in local IL-6 levels.

How was the proper sample size determined in each protocol of the study? The figure legend 1 indicates that 66 controls, 67 susceptible and 50 resilient mice were used in the chronic social defeat stress paradigm. On the other hand, the immunohistochemistry data that strongly support the assumptions of the study were obtained from only 2 to 3 mice per group, according to the Material and methods section. Such small n for the histology studies does not generate confidence in the results.

It is important to highlight here that the experimental design required the generation of very large cohorts of mice to be exposed to the social defeat stress paradigm and then further classified as susceptible or resilient- based on the behavioral response. For each follow up analysis we then compared susceptible and resilient mice from each cohort. Due to the nature of the behavioral experiments and the reproducible results in the additional follow up analyses (e.g. qPCR, immunohistochemistry on several brain regions with various antibodies, and electron microscopy), we used a large number of mice for the initial assessment and then repeated the experiments in several cohorts.

All graphs in the results, particularly for the behavior assessment, should show the distribution of each individual sample, as points in the bar graph.

Bar graphs have been replaced with individual data points.

For rigor and reproducibility, the time of the day for the behavioral assessments and the anesthetic used for the stereotaxic surgery need to be added to the Material and methods section.

The information has been included in the Materials and methods Section.

How did the authors confirm that lysolecithin injection hit the desired site in the mPFC? What is the success rate of the stereotaxic surgery?

The identification of the injection site was validated by histological analysis. Because the injection sites based on stereotaxic coordinates were only visible at 7dpi, we confirmed the injection in the mPFC in five mice that were sacrificed at 7dpi. In the subsequent cohorts, we collected mice only at 21dpi in order to follow them up longitudinally and assess the social behavior on the same mouse at 7dpi and 21dpi.

Were the same mice tested for social interaction at 7- and 21- days post lysolecithin injection? Is it possible that those mice present any adaptation to the experimental protocol that could bias the results of social interaction?

Yes. The same mice were tested for social interaction at 7dpi and 21 dpi so that we could directly compare the social behavior at the time of demyelination and after remyelination. It is theoretically possible that mice could have developed some adaptation to the experimental protocol, but it is quite unlikely based on our previous experience.

The images in the Figure 4B do not clearly show an increase of MBP expression in mice 21 days after LPC injection, compared to 7 days post treatment. Higher magnification images might better represent the remyelination process after 21 days.

We purposely selected low magnification images to show MBP immunoreactivity within a larger area, because the temporal window of LPC-induced demyelination and remyelination has been well characterized in several previous studies. In addition, based on the comments of the other reviewers, we examined internodal length at higher magnification and demonstrated indistinguishable internodal length after remyelination.

Because IL-6 circulating levels or mRNA expression were not assessed, the discussion on IL-6 role in resilience and susceptibility to stress seems unwarranted.

The circulating IL-6 level before and after social defeat has been assessed in previous publications (e.g. Hodes et al., 2014). The local *Il6* mRNA level has been assessed and presented in the previous version of the manuscript but was removed due to the length limit of a short report and replaced with LPC experiments suggested by the editor. However, we believe this is an important aspect that could potentially explain the region-specific differences in oligodendrocyte response observed in the mPFC and NAc. Therefore, we kept it in the Discussion.

Reviewer #2:

[…] Overall, this is an exciting study demonstrating the role of myelin in social behaviour and maladaptive social behaviour following stress conditions in functionally relevant brain regions. The authors present evidence that resilience to stress requires maintenance of healthy myelination in mPFC, and that a susceptible subject exhibits impaired oligodendrocyte differentiation dynamics following the stressful condition specifically in mPFC, implicating disordered mPFC myelination in susceptibility to stress-induced social withdrawal. Together, the manuscript represents an important contribution to the field and lends compelling support for the concept that dysmyelination can play important roles in vulnerability to stress, and mood disorders.I have the following suggestions for improvement to better support the interesting claims:• Regarding statistical rigor: Throughout the text, authors should give exact definition of "n" and state when it is number of mice, number of axons etc. Also, "imaging volumes" should be defined.

These concerns have been addressed in the response to the editor.

• Figure 1A. The social defeat paradigm could be explained better by including the social interaction test that is performed in the end of chronic social defeat setting. This will make it clearer for the reader and it would be more coherent for Figure 4, and also help explain Figure 1B which shows the time spent in interaction zone that is measured during social interaction test.• Figure 1B. Including social interaction ratio as well will make it more consistent with the rest of the figure.

We have included additional description and citations of how the social interaction tests were performed in the Materials and methods section. We have also revised Figure 1 to include additional panels (Figure 1B and Figure 1C) to help make the manuscript consistent when talking about the social interaction test.

• Including more information on control group animals in the text would make it easier to understand the differences between mice groups. For example, "control group, mice that were not exposed to any aggressor" would be helpful in understanding the comparisons.

Additional information is included.

• Throughout the study, authors use MBP immunohistochemistry alone to measure the myelin internode length. However, MBP alone may not differentiate the difference between an internode and an axon going into the field in z-axis. Co-staining with caspr should clarify that distinction. Rather than re-analyze all of the internode data this way, the authors could demonstrate in a sample group that caspr co-staining validates the original internode quantifications.

We have performed additional experiments to quantify the internodal length using co-staining of Caspr and MBP. These results have been included in the revised Figure 2.

• Including a scatter plot (g-ratio vs axon calibre) is a more informative way to present g-ratio data.

Bar graphs have been replaced with scatter plots to present g-ratio.

• What does% MBP+ area measurement represent? Unaffected N.Ac. shows reduction in susceptible and resilient mice, but the affected mPFC show no difference for% MBP+ area. Maybe there are fewer myelinated axons in N.Ac for susceptible and resilient mice groups, does the EM data give any suggestion towards this?

The% MBP+ area reflects total amount of MBP in the region. It is possible that there were fewer myelinated axons in the NAc, however the overall content was not quantified.

• Figure 4. Regarding social interaction test, presentation of time spent in the interaction zone or distance travelled in the presence and absence of a target mouse would show that subject mouse is fine to leave the interaction zone (i.e. motor abilities). In other words, unpacking the social interaction ratio.

We did not include the results of total distance traveled in the manuscript, although we did not identify any differences in the total distance traveled among all groups.

• Figure 4. At 21dpi, authors show that there is recovery of social interaction with the recovery of myelination in mPFC. Does the recovery of internode length also occur? Or MBP area, to reconcile with the initial mPFC phenotype they show in Figure 2?

Yes, we performed additional experiments and quantified the internodal length using co-staining of Caspr and MBP. The internodal length was indistinguishable between the control and LPC group.

Minor Comments:• The reduction in H3k9me3 immunostaining in Olig2+ cells in mPFC is consistent with the observed decreased OPC differentiation, but it seems an overstatement to claim "region-specific epigenetic dysregulation of oligodendrocyte differentiation", There is not at present evidence that the lack of mPFC oligodendrogenesis is primarily due to epigenetic dysregulation…the reduced H3K9me3 could alternatively simply reflect the loss of differentiation signal from the environment. Wording should be modified.

We have revised text accordingly.

• Regarding wording, some softening of claims should be made in the Discussion. For example, "social avoidance behavior can be detected after chronic social defeat stress as well as after focal demyelination in the mPFC, and is therefore caused by hypomyelination." Would be better reading, "…and therefore can be caused by hypomyelination"

We have revised the text to incorporate the comments.

Reviewer #3:

[…] The data are potentially interesting. At the same time I have a number of questions and concerns, which I believe the authors should address:a) It would be helpful if the authors would plot individual data points in the bar graphs to better estimate group size and distribution of the data. How exactly where resilient and vulnerable animals defined?

This point has been addressed above.

b) The first experiment included a large number of mice, while all follow-up analyses only used very few animals per group (e.g. 3 or 4). How were these animals selected from the respective groups? What was the social avoidance behavior of the selected mice? Specifically I am worried that if only extremes of a group were selected that the results would not be representative of the whole group.

This point has been addressed above.

c) For the correlation analyses in Figure 1F and 2F: Are animals from all 3 groups included? The data would be more convincing if a correlation of intermodal length and social interaction ratio would be present within experimental groups or at least within the CSD group.

In response to this comment, we have revised Figure 1 and Figure 2 by separating the correlation in control and defeated groups.

d) The differences in histone modification are interesting, but only correlative. The authors should avoid claiming a causal relationship with myelination or stress susceptibility.

This point has been addressed in response to Editor’s point.

e) I am not convinced by the conclusions the authors draw from the LPC experiment. At 7 days following treatment, the authors observed a reduction in MBP levels. However, MBP levels were not significantly different between stress resilient and susceptible mice. The manipulation does therefore not reflect the stress-induced situation, even though a similar behavioral phenotype was observed.

This point has been addressed above.

f) How specific is the social avoidance phenotype induced by focal demyelination using LPC? The authors would need to show that the animals are not generally impaired and healthy.

The altered social preference behavior was only observed in mice analyzed seven days after focal LPC injection. Although not shown, mice were not generally motor impaired or unhealthy, as also supported by the total distance traveled during the behavioral tests.

g) While the final experiment parallels the behavioral effect in stress susceptible mice (but see the issue with MBP levels), it does not indicate causality. For that, the authors would need to show that a prevention of demyelination would increase stress resilience following CSD.Overall, I think the data are novel and interesting. However, the authors overstate their conclusions, as all observations are correlational and no experiments were performed that would indicate a causal relationship between the observed differential myelination phenotype of resilient and susceptible animals with their social behavior or epigenetic regulation.

We thank the reviewer for finding our data novel and interesting. We agree with the reviewer that the LPC-induced demyelination is not equivalent to social defeat stress-induced hypomyelination, as only the former one induced toxicity. We have clearly addressed this point in our discussion. However, the LPC experiment was the most direct way to manipulate myelin content and demonstrate a link between myelinated segments in the mPFC and social behavior. We have revised the text in order to avoid overstating our conclusions.